# Development of Procathepsin L (pCTS-L)-Inhibiting Lanosterol-Carrying Liposome Nanoparticles to Treat Lethal Sepsis

**DOI:** 10.3390/ijms24108649

**Published:** 2023-05-12

**Authors:** Weiqiang Chen, Cassie Shu Zhu, Xiaoling Qiang, Shujin Chen, Jianhua Li, Ping Wang, Kevin J. Tracey, Haichao Wang

**Affiliations:** 1The Feinstein Institutes for Medical Research, Northwell Health, 350 Community Drive, Manhasset, New York, NY 11030, USA; wchen6@northwell.edu (W.C.); czhu3@northwell.edu (C.S.Z.); xqiang@northwell.edu (X.Q.); sarahchen2024@gmail.com (S.C.); jli@northwell.edu (J.L.); pwang@northwell.edu (P.W.); kjtracey@northwell.edu (K.J.T.); 2Donald and Barbara Zucker School of Medicine at Hofstra/Northwell, 500 Hofstra Blvd., Hempstead, New York, NY 11549, USA

**Keywords:** sepsis, innate immune cells, procathepsin-L, lanosterol, liposome, inflammation

## Abstract

The pathogenesis of microbial infections and sepsis is partly attributable to dysregulated innate immune responses propagated by late-acting proinflammatory mediators such as procathepsin L (pCTS-L). It was previously not known whether any natural product could inhibit pCTS-L-mediated inflammation or could be strategically developed into a potential sepsis therapy. Here, we report that systemic screening of a NatProduct Collection of 800 natural products led to the identification of a lipophilic sterol, lanosterol (LAN), as a selective inhibitor of pCTS-L-induced production of cytokines [e.g., Tumor Necrosis Factor (TNF) and Interleukin-6 (IL-6)] and chemokines [e.g., Monocyte Chemoattractant Protein-1 (MCP-1) and Epithelial Neutrophil-Activating Peptide (ENA-78)] in innate immune cells. To improve its bioavailability, we generated LAN-carrying liposome nanoparticles and found that these LAN-containing liposomes (LAN-L) similarly inhibited pCTS-L-induced production of several chemokines [e.g., MCP-1, Regulated upon Activation, Normal T Cell Expressed and Presumably Secreted (RANTES) and Macrophage Inflammatory Protein-2 (MIP-2)] in human blood mononuclear cells (PBMCs). In vivo, these LAN-carrying liposomes effectively rescued mice from lethal sepsis even when the first dose was given at 24 h post the onset of this disease. This protection was associated with a significant attenuation of sepsis-induced tissue injury and systemic accumulation of serval surrogate biomarkers [e.g., IL-6, Keratinocyte-derived Chemokine (KC), and Soluble Tumor Necrosis Factor Receptor I (sTNFRI)]. These findings support an exciting possibility to develop liposome nanoparticles carrying anti-inflammatory sterols as potential therapies for human sepsis and other inflammatory diseases.

## 1. Introduction

Microbial infections and the resultant sepsis syndromes are the most common causes of death in intensive care units, accounting for approximately 20% of total deaths worldwide [1]. Its pathogenesis is partly attributable to dysregulated innate immune responses (e.g., hyperinflammation and immunosuppression) to lethal infections orchestrated by early and late proinflammatory cytokines [2]. For instance, upon innate recognition of “pathogen-associated molecular patterns” (PAMPs, e.g., bacterial endotoxins, lipopolysaccharide (LPS)) by respective pattern recognition receptors (PRRs, e.g., the Toll-like receptor 4 (TLR4)) [3], macrophages and monocytes sequentially produce and/or release “early” cytokines (e.g., tumor necrosis factor (TNF) and interleukin-1β (IL-1β)) [4,5,6], pathogenic chemicals (e.g., lactate) [7,8,9], as well as late-acting proinflammatory mediators such as high mobility group box 1 (HMGB1) [10] and procathepsin-L (pCTS-L) [11]. In contrast to early proinflammatory cytokines, these late-acting mediators can be pharmacologically targeted in a delayed fashion [11,12], thereby providing relatively wider therapeutic windows for lethal sepsis. It is thus important to search for small molecule natural products capable of inhibiting pCTS-L-mediated dysregulated inflammation for the clinical management of human sepsis.

Natural products refer to all primary and secondary metabolites of microbes, plants, and animals that have been evolutionarily selected over billions of years for adaptive interactions with various biomolecules [13]. Traditionally, these natural products serve as a major source of lead chemicals for the contemporary discovery and development of many pharmaceutical drugs [13,14]. However, the search for lead compounds from natural products is often time-consuming and labor-intensive, thereby hampering the efficient identification of molecular targets for most natural products [14]. In the present study, we sought to develop a high-throughput screening assay to search for small molecule inhibitors of pCTS-L-mediated inflammation in order to develop novel therapies for lethal sepsis. Here, we provided compelling evidence to support: (i) a natural lipophilic sterol, lanosterol (LAN), as a selective inhibitor of pCTS-L-induced inflammation; and (ii) a novel LAN-carrying liposome nanoparticles as a promising therapy for lethal sepsis.

## 2. Results

### 2.1. Identification of Lanosterol as a Selective Inhibitor of pCTS-L-Induced TNF Secretion

To search for small molecule pCTS-L inhibitors, we developed a semi-high-throughput macrophage cell-based bioassay to screen a NatProduct Collection of 800 compounds for potential capacities to suppress pCTS-L-induced TNF production (Figure 1A). The experimental conditions were optimized by titrating macrophage cell densities (to 80–90% confluence) and pCTS-L concentrations (to 1.0 μg/mL), and the pCTS-L-induced TNF secretion was normalized by the percentage of change between negative (“−pCTS-L plus DMSO vehicle”) and positive (“+pCTS-L alone”) controls. A systemic screening of 800 natural products of the NatProduct Collection led to the identification of three lead compounds (i.e., 3-hydroxyflavone, theaflavin, and lanosterol; Figure 1B) that selectively inhibited TNF secretion induced by pCTS-L, but not LPS. One lead compound, lanosterol (LAN), dose-dependently inhibited pCTS-L-induced TNF production in murine macrophages with an estimated IC_50_ around 20.0 µM (Figure 1B,C).

### 2.2. LAN Significantly Suppressed pCTS-L-Induced Production of TNF and Other Chemokines in Primary Human Peripheral Blood Mononuclear Cells (PBMCs)

To confirm its pCTS-L-inhibiting properties, we solubilized LAN in ethanol and then examined its effect on pCTS-L-induced production of various cytokines/chemokines in primary human PBMCs. Consistent with our previous report [11], recombinant human pCTS-L effectively induced several cytokines (e.g., TNF and IL-6) and chemokines [e.g., MCP-1, ENA-78/LIX, and Growth-Regulated Oncogene (GRO)] in human PBMCs (Figure 2A,B). However, the pCTS-L-induced TNF secretion was almost completely abrogated by the co-addition of LAN at a relatively high concentration (40 μM, Figure 2A,B). Similarly, the pCTS-L-induced production of other cytokines (e.g., IL-6) and chemokines (e.g., ENA-78 and MCP-1) was also markedly inhibited by the co-addition of LAN (Figure 2B). Consistent with previous reports [15], bacterial endotoxin markedly induced several cytokines (e.g., TNF and IL-6) and chemokines (MCP-1, ENA-78, and GRO) in human PBMCs (Figure 2A,B). However, at the concentration that significantly attenuated pCTS-L-induced inflammation, LAN still failed to reduce LPS-induced production of any cytokines and chemokines (Figure 2A,B), confirming that LAN selectively attenuated the pCTS-L-mediated inflammatory response.

### 2.3. LAN-Carrying Liposomes Similarly Suppressed pCTS-L-Induced Inflammation

In order to assess the therapeutic efficacy of this lipophilic sterol in vivo, we generated phospholipid-based liposomes carrying either lanosterol (LAN-L) or cholesterol (CHO-L) (Figure 3A) using phospholipids with long and saturated acyl chains (such as 1,2-dipalmitoyl-sn-glycero-3-phosphatidylcholine, DPPC) that can produce relatively more stable liposomes. As lipophilic sterols, both LAN and CHO could be encapsulated entirely into the bilayer membrane of DPPC liposomes with average particle sizes of 423.07 nm and 112.42 nm, respectively (Figure 3A). To confirm the pCTS-L-inhibiting activities of lanosterol, the DPPC-based liposomes containing either LAN (LAN-L) or CHO (CHO-L) were added to macrophage cultures to examine their impact on the pCTS-L-induced production of cytokines/chemokines. In contrast to liposomes carrying a control sterol, cholesterol (CHO-L), liposomes containing LAN (LAN-L) significantly inhibited pCTS-L-induced production of Granulocyte-Macrophage Colony-Stimulating Factor (GM-CSF) and sTNFRI/II, as well as several chemokines (MCP-1, MIP-2, and RANTES) in macrophage cultures (Figure 3B), confirming that DPPC-based liposomes containing LAN maintained its pCTS-L-inhibiting properties.

### 2.4. LAN-Carrying Liposome Rescued Mice from Lethal Sepsis

To evaluate the LAN’s therapeutic potential, we administered DPPC-based liposomes containing either LAN (LAN-L) or CHO (CHO-L) in a delayed fashion: starting at 24 h after onset of sepsis, a time point when circulating pCTS-L approached plateau concentrations and some animals started to succumb to death in experimental sepsis [11]. As an experimental control, CHO-L did not significantly affect animal survival when given intraperitoneally at 24 h and 48 h post CLP (Figure 4A). In sharp contrast, the DPPC-based liposomes carrying LAN (LAN-L) significantly rescued animals from lethal sepsis even when the first dose was given at 24 h post the onset of sepsis (Figure 4A). Based on their different molecular weights (LAN = 426.72 Daltons; CHO = 386.55 Daltons), the calculated molar concentrations of CHO (14.5 mg/kg) and LAN (16.0 mg/kg) intraperitoneally given to different groups of animals were virtually comparable (i.e., >37.5 μM).

To understand the protective mechanisms of LAN-L, we examined its effect on sepsis-induced tissue injury and systemic inflammation. Consistent with our previous reports [11,16], sepsis induced marked liver injuries as judged by sinusoidal congestion (arrows), vacuolization of hepatocyte cytoplasm, and parenchymal necrosis (Figure 4B,C). In sharp contrast to cholesterol-containing liposomes (CHO-L), which did not affect sepsis-induced liver injury (Figure 4B,C), LAN-L significantly attenuated sepsis-induced liver injury (Figure 4B,C). Consistent with our previous findings [11,16,17,18], sepsis also induced a marked elevation of blood levels of G-CSF, IL-6, KC, and sTNFR1 (Figure 4D) at 24 h post CLP. In contrast to CHO-L, LAN-L significantly suppressed sepsis-induced elevation of IL-6, KC, and sTNFRI (Figure 4D), suggesting that LAN-carrying liposomes (LAN-L) conferred protection against lethal sepsis partly by attenuating sepsis-induced tissue injury and dysregulated inflammation.

## 3. Discussion

Lacking effective therapies for sepsis other than adjunctive use of antibiotics, fluid resuscitation, and supportive care [19], it is still urgent to find small molecule inhibitors for late-acting pathogenic mediators that may offer relatively wider therapeutic windows. Here, we demonstrated that a lipophilic sterol, lanosterol, selectively attenuated pCTS-L-induced production of proinflammatory cytokines (e.g., TNF and IL-6) and chemokines (e.g., ENA-78, MCP-1, MIP-2, or RANTES) in murine macrophages or human PBMCs. Consequently, LAN-containing liposomes rescued mice from lethal sepsis even when the first dose was given in a delayed regimen (i.e., 24 h post the onset of sepsis). Our findings mirrored previous observations that a cis-trans isomer of lanosterol, euphol (Figure 5), conferred protection against experimental colitis [20], autoimmune encephalomyelitis [21], carrageenan-induced paw edema [22], and 12-O-tetradecanoylphorbol-13-acetate (TPA)-induced skin dermatitis [23]. Furthermore, our current findings were also in agreement with a recent report that pharmacological or genetic elevation of lanosterol accumulation in macrophages led to an attenuated cytokine production and endotoxemic animal lethality [24], supporting an important role for lanosterol in the negative regulation of innate immunity. Collectively, we and others have provided compelling evidence to support the traditional use of lanosterol-containing Goji Berries (*Lycium barbarum*) [25] and euphol-abundant Fire Stick (*Euphorbia tirucalli*) [23,26] in the treatment of various inflammatory ailments that include pneumonia, asthma, and cough (Figure 5).

In previous animal studies, the cis-trans isomer of lanosterol, euphol, was dissolved either in nonionic surfactant (e.g., 5% Tween 80) [23] or organic solvent (e.g., mixture of acetone/ethanol at a ratio of 3:1) [20,21,22] and was subsequently given to animals either orally (by gavage) [23] or topically on the skin [20,21,22], respectively. In light of the notion that plant phytosterols are often incorporated into micelles of diet fats [27] to increase their bioavailability in vivo [28], we generated DPPC-based liposomes to encapsulate lanosterol to facilitate their bioavailability. Regardless of whether LAN was dissolved in organic solvent (ethanol) or encapsulated in DPPC-based liposomes, it similarly inhibited pCTS-L-induced production of several proinflammatory cytokines (e.g., TNF and IL-6) and chemokines (e.g., ENA-78, MCP-1, MIP-2, or RANTES) in murine macrophages or human PBMCs.

The molecular mechanisms underlying the LAN-mediated inhibition of pCTS-L-induced inflammation remain an exciting subject for future investigations. Interestingly, it is known that liposomes can directly interact with macrophage/monocyte cell surface components, and relatively large liposomes (>100 nm in diameters) can even be endocytosed by resident macrophages in the liver and spleen [29,30,31,32]. Given the critical role of TLR4 and RAGE in pCTS-L-induced production of IL-6, ENA-78/LIX, RANTES, MCP-1, and MIP-2/GRO-β [11], it will be important to investigate whether LAN directly interacts with pCTS-L or its cell surface receptors (e.g., TLR4 and RAGE) to interfere with respective ligand/receptor interactions. This is possible because other pCTS-L inhibitors (e.g., pCTS-L-neutralizing monoclonal antibodies) attenuated pCTS-L-induced production of similar chemokines (e.g., ENA-78/LIX, RANTES, MCP-1, GRO-β/MIP-2) by impairing pCTS-L interaction with TLR4 and RAGE receptors [11]. Similarly, it is not known whether LAN-containing liposomes can fuse with macrophage/monocyte cytoplasmic membranes, enabling lanosterol interaction with the saturated alkyl chains of the phospholipids and sphingolipids in “lipid rafts”—dynamic membrane domains harboring various cell surface receptors (e.g., TLR4 and RAGE). Finally, it is not known whether LAN-containing liposomes are engulfed by macrophages, thereby elevating macrophage lanosterol contents to inhibit pCTS-L-induced inflammation via other intracellular signaling molecules. This is likely because an elevated accumulation of lanosterol in innate immune cells has been linked to an attenuated cytokine production and an improved outcome from lethal endotoxemia in preclinical settings [24].

Because liposome-based therapies may suffer from physical and chemical instabilities [31], lipophilic sterols such as CHO and LAN can be strategically employed to increase the membrane stability of phospholipid-based liposomes [33,34,35]. Indeed, both sterols can readily incorporate into bilayer of amphipathic DPPC phospholipids, thereby intimately interacting with the long and saturated acyl chains of DPPC to increase liposome stabilities [29,33]. Harboring two additional methyl groups on the otherwise flat alpha face of cholesterol (Figure 3A), LAN has a slightly lower capacity to condense the acyl chains of DPPC phospholipids to increase stability of DPPC bilayers [36]. Nevertheless, the anti-inflammatory LAN can still be employed to increase the membrane stability of liposomes [33,34,35] and may be pharmacologically useful for generating liposomes to deliver other lipophilic (in the lipid bilayer) or hydrophilic immune-modulating drugs (inside the internal aqueous pockets).

To our knowledge, the DPPC-based liposome technology has not yet been employed to test the therapeutic efficacy of lanosterol in animal models of sepsis. However, another cyclodextrin-based nanotechnology that strategically employs the hydrophobic doughnut-shaped pockets of a class of cyclic oligosaccharides consisting of 6, 7, or 8 glucopyranose to capture lanosterol has been developed in an attempt to treat cataracts in animals [37]. At present, the mechanism underlying LAN-mediated protection remains elusive, but was at least partly attributable to its inhibition of pCTS-L-induced tissue injury and dysregulated inflammation. In vivo, DPPC-based liposomes containing LAN, but not CHO, significantly attenuated sepsis-induced liver injury and systemic accumulation of IL-6, sTNFRI, and KC, three pCTS-L-inducible cytokines/chemokines known as surrogate markers of experimental sepsis [38,39,40]. Given the pathogenic role of pCTS-L in pancreatitis [41], atherosclerosis [42], renal disease [43], vascular intimal hyperplasia [44], arthritis [45], and colitis [46], it will also be important to test the therapeutic efficacy of LAN-containing liposomes in other inflammatory diseases.

There are a number of limitations in the present study: (i) It is not known whether LAN-containing liposomes can be given orally to achieve similar protection against lethal sepsis. (ii) We do not know whether LAN-containing liposomes are also protective against other infectious diseases such as COVID-19, because pathogenic up-regulation of CTS-L might facilitate the entry of SARS-CoV-2 into host cells [47]. (iii) We have not yet investigated whether LAN can directly bind pCTS-L or its receptors (e.g., TLR4 or RAGE) particularly because of technical limit of currently available SPR technologies in assessing protein interaction with hydrophobic molecules. (iv) Although we purposely used phospholipids with long and saturated acyl chains to produce liposomes with relatively higher stability, liposomes commonly suffer from storage instability caused by particle fusion and aggregation, as well as phospholipid hydrolysis or oxidation. That was why as soon as the LAN- or CHO-carrying liposomes were generated, we immediately conducted all cellular and animal experiments within 1–6 weeks, long before the subsequent appearance of precipitates (possibly due to liposome self-aggregation, coalescence, and flocculation) after 6–9 months of prolonged storage (4 °C). Therefore, stability studies for the lanosterol-carrying liposomes are critically needed before conducting future clinical trials in accordance with quality guidelines of the International Council for Harmonisation of Technical Requirements for Pharmaceuticals for Human Use (ICH).

To overcome liposomes’ instability issues, water-soluble carriers (e.g., anhydrous carbohydrates such as dextrose, mannitol, sorbitol, maltodextrin, and microcrystalline cellulose) can be employed to produce a form of solid dry and free-flowing granular phospholipid liposomes, “proliposomes”, which directly generate isotonic liposomal dispersion upon contacting water or any biological fluid [48]. The superior stability of proliposomes makes them more suitable for delivery of liposomes with poor oral [49] or nasal [50] stability and bioavailability. For instance, paclitaxel-loaded proliposomes are currently being tested in clinical trials for patients with low-grade thyroid or bladder cancer [51], providing hope for future nanoparticle-based research. Therapeutic proliposomes that are formulated with surfactant-like phospholipids (e.g., DPPC) are particularly hopeful to offer controlled release and enhanced stability of anti-inflammatory agents in pulmonary nanomedicine [50]. Despite the many limitations of the present study, our discovery of LAN as a selective inhibitor of pCTS-L-induced dysregulated inflammation has suggested an exciting possibility of developing novel liposome nanoparticles to fight against lethal sepsis. It will thus be important to develop LAN-carrying liposomes or proliposomes containing lung surfactant-like phospholipids (e.g., DPPC) and translate these pre-clinical findings into the clinical management of pulmonary and other systemic infections.

## 4. Materials and Methods

### 4.1. Cell Culture

Murine macrophage-like RAW 264.7 cells were obtained from ATCC. Human blood was purchased from the New York Blood Center (Long Island City, NY, USA), and human peripheral blood mononuclear cells (PBMCs) were isolated via density gradient centrifugation through Ficoll (Ficoll-Paque PLUS) as previously described [11,52,53,54]. Murine macrophages and human PBMCs were cultured in DMEM supplemented with 1% penicillin/streptomycin and 10% FBS or 10% human serum. When they reached 80–90% confluence, adherent cells were gently washed with, and immediately cultured in, OPTI-MEM I before stimulating with crude LPS (*E. coli* 0111:B4, #L4130, Sigma-Aldrich, St. Louis, MO, USA) or recombinant human or murine pCTS-L in the absence or presence each of the 800 compounds in the NatProduct chemical library (supplied in DMSO at 10 mM), lanosterol (#L5768, Sigma-Aldrich), or lanosterol-containing liposomes. For cellular studies, lanosterol was also dissolved in ethanol (5 mg/mL) or encapsulated in liposomes as described below. The extracellular concentrations of various other cytokines/chemokines were determined using ELISA, Western blotting, or Cytokine Antibody Arrays as previously described [11,52,53,54].

### 4.2. Preparation of Recombinant Human and Murine pCTS-L Proteins

As previously described [11], recombinant human and murine pCTS-L proteins corresponding to residue 17-333 or 18-334 of respective precathepsin L with a N-histidine tag were expressed in *E. coli* BL21 (DE3) pLysS cells. After sonication to disrupt the bacteria, the pCTS-L-containing inclusion bodies were isolated by differential centrifugation following extensive washing in 1 × PBS containing 1% Triton X-100. The inclusion bodies were then solubilized in 8 M urea and refolded via dialysis in 10 mM Tris buffer (pH 8.0) containing N-lauroylsarcosine. Subsequently, the recombinant pCTS-L protein was then further purified by histidine-affinity chromatography, followed by extensive Triton X-114 extractions to remove contaminating endotoxins. Recombinant pCTS-L protein was tested for LPS content using the chromogenic *Limulus* amebocyte lysate assay (Endochrome; Charles River, Wilmington, MA, USA), and the endotoxin content was less than 0.01 U per microgram of recombinant protein.

### 4.3. High-Throughput Screening of Natural Product Chemical Library for pCTS-L Inhibitors

We obtained a NatProduct Collection of 800 pure natural products and their derivatives (supplied at 10 mM in DMSO) from the MicroSource Discovery System Inc. and used it to screen for potential pCTS-L inhibitors using a semi-high-throughput cell-based bioassay on 96-well plates. Briefly, murine macrophage-like RAW 264.7 cells were stimulated with pCTS-L either alone or in the presence of each chemical at several concentrations for 16 h, and levels of TNF in the cell-conditioned culture medium was determined using TNF DuoSet ELISA kit (Cat# DY410, R&D Systems, Minneapolis, MN, USA) according to the manufacturer’s instructions. Both positive (“+pCTS-L alone”) and negative (“−pCTS-L”) controls were used to normalize TNF secretion data (as %) from different plates within and between experiments.

### 4.4. Western Blotting

Human PBMCs were stimulated with LPS or recombinant human pCTS-L for 16 h, and the level of TNF in human PBMC-conditioned culture medium was determined using Western blotting analysis using commercial goat anti-human TNF polyclonal antibodies (Cat. # SC-1350, Santa Cruz, CA, USA). Equal volume of cell-conditioned culture medium was resolved on sodium dodecyl sulfate (SDS)-polyacrylamide gels and transferred to polyvinylidene difluoride (PVDF) membranes. After blocking with 5% nonfat milk, the membranes were incubated with the appropriate antibodies (anti-TNF, 1:1000) overnight. Subsequently, the membranes were incubated with the appropriate secondary antibodies, and the immune-reactive bands were visualized by chemiluminescence. The relative TNF levels were determined using the UN-SCAN-IT Gel Analysis Software Version 7.1 (Silk Scientific Inc., Orem, UT, USA), and expressed in arbitrary units (AU).

### 4.5. Generation of Lanosterol- or Cholesterol-Carrying Liposomes

We employed a thin-film hydration method to produce lanosterol (LAN)- (#L5768, Sigma-Aldrich) or cholesterol (CHO)- (#C3045, Sigma-Aldrich) carrying liposomes by mixing 1,2-dipalmitoyl-sn-glycero-3-phosphocholine (DPPC) with LAN and CHO at a molar ratio of 50–50%. Briefly, the LAN/DPPC or CHO/DPPC mixtures were dissolved in chloroform in round-bottom flasks, and the solvent was then removed under vacuum in a rotary evaporator. The lipid film was then mechanically scraped off with a spatula, and an appropriate amount of saline solution was added to the round bottom flask to give a final concentration of 15 mM solution. The LAN/DPPC or CHO/DPPC lipid sheets were then converted into liposomes via sonication using a micro homogenizer tip (MISONIX Fisher Scientific Sonicator Ultrasonic Processor XL) until the solution appeared homogenous (~5 min). Dynamic light scattering (DLS) was used to determine liposomes’ size in the sub-micron range by using light scattering from a laser that passes through liposomal solution to analyze the intensity of scattered light as a function of time. The liposome suspensions were then incubated in a 60 °C water bath for 35 min and then extruded using an “Avanti” mini-extruder (Avanti Polar Lipids, Alabaster, AL, USA) five times through a 1.0 μm polycarbonate membrane, followed by five times through a 0.4 μm membrane. The extruded liposomes containing lanosterol or cholesterol trapped in the phospholipid bilayer were then tested in animal models of sepsis.

### 4.6. Cytokine Antibody Arrays

Murine Cytokine Antibody Arrays (Cat.# AAM-CYT-3-8, RayBiotech Inc., Norcross, GA, USA), which simultaneously detect 62 cytokines on 1 membrane, were used to measure relative cytokine concentrations in macrophage-conditioned culture medium or murine serum as described previously [11,52,53]. Human Cytokine Antibody C3 Arrays (Cat.# AAH-CYT-3-8), which detect 42 cytokines on 1 membrane, were used to determine cytokine concentrations in human PBMC-conditioned culture medium as previously described [11,52,53,54].

### 4.7. Animal Model of Lethal Sepsis

This study was conducted in accordance with policies of the NIH Guide for the Care and Use of Laboratory Animals and approved by the IACUC (protocol # 2017-003 Term II, approved on 28 April 2020) of the FIMR. Balb/C mice (male and female, 7–8 weeks old, 20–25 g) were subjected to lethal sepsis by a surgical procedure termed as cecal ligation and puncture (CLP) as previously described [11,52,53,54]. Briefly, the cecum of Balb/C mice was ligated at 5.0 mm from the cecal tip, and then punctured once with a 22-gauge needle. To alleviate surgical pain, all animals were given a dose of anesthetics (e.g., buprenorphine, 0.05 mg/kg, s.c.) immediately prior to CLP surgery, and small amounts of anesthetics (such as bupivacaine and lidocaine) was injected locally around the incision site immediately after the closure of the abdominal wound of CLP surgery. At 30 min after CLP, all animals were given a subcutaneous dose of imipenem/cilastatin (0.5 mg/mouse) (Primaxin, Merck & Co., Inc., Rahway, NJ, USA) and resuscitation with normal sterile saline solution (20 mL/kg). For the pre-clinical study, animals were randomly assigned to different experimental groups, and DPPC-based liposomes carrying either lanosterol (LAN-L) or cholesterol (CHO-L) were intraperitoneally administered to septic mice at 24 h and 48 h post CLP, and animal survival rates were monitored for up to two weeks.

### 4.8. Tissue Injury

The liver tissue was harvested at 24 h post CLP and fixed in 10% buffered formalin before being embedded in paraffin. Paraffin-embedded tissues were cut into 5 μm sections, stained with hematoxylin-eosin, and examined under light microscopy. As previously described [11,55], liver parenchymal injury was assessed in a blinded fashion by the sum of three different Suzuki scores ranging from 0 to 4 for sinusoidal congestion, vacuolization of hepatocyte cytoplasm, and parenchymal necrosis. Using a weighted equation with a maximum score of 100 per field, the parameter scores were calculated and then averaged as the final liver injury score in each experimental group.

### 4.9. Statistical Analysis

All data were assessed for normality using the Shapiro–Wilk test before conducting appropriate statistical tests among groups. The comparison of two independent samples was assessed using Student’s t test and the Mann–Whitney test for Gaussian and non-Gaussian distributed samples, respectively. For comparison among multiple groups with normal data distribution, the differences were analyzed using one-way analyses of variance (ANOVA) followed by the Fisher Least Significant Difference (LSD) test. For comparison among multiple groups with non-normal (skewed) distribution, the statistical differences were evaluated with the non-parametric Kruskal–Wallis ANOVA test. For survival studies, the Kaplan–Meier method was used to compare the differences in mortality rates between groups with the nonparametric log-rank post hoc test. A *p* value < 0.05 was considered statistically significant.

## Figures and Tables

**Figure 1 ijms-24-08649-f001:**
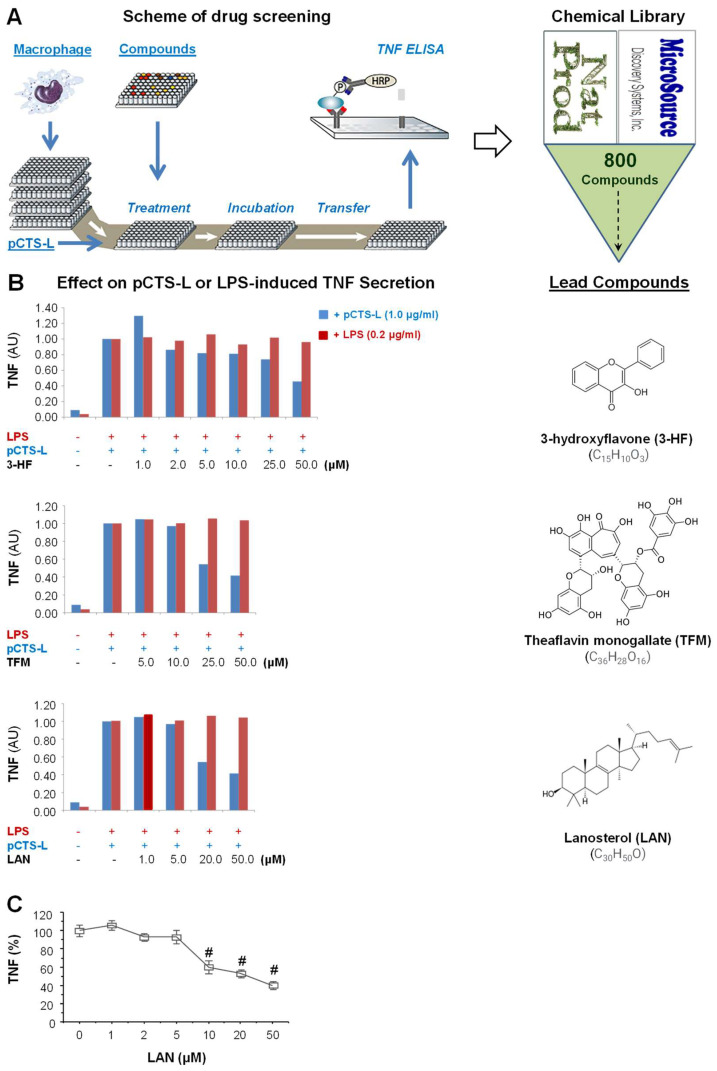
High-throughput screen of a chemical library of 800 natural products for inhibitors of pCTS-L-induced TNF secretion. (**A**) Scheme for the high-throughput screening of pCTS-L inhibitors. Murine macrophage-like RAW 264.7 cells were cultured on 96-well plates until 80–90% confluence and were stimulated with recombinant murine pCTS-L (1.0 μg/mL) or LPS (0.2 μg/mL) in the absence or presence of each of the 800 compounds for 16 h, and levels of TNF in cell-conditioned medium were determined using murine TNF ELISA. (**B**,**C**) Effects of three lead compounds on pCTS-L-induced TNF secretion by murine macrophages. Note that all three lead compounds dose-dependently inhibited TNF secretion induced by pCTS-L, but not LPS. Panel C showed the average TNF suppression (%) of three independent experiments (*n* = 3). #, *p* < 0.05 versus “+pCTS-L alone”.

**Figure 2 ijms-24-08649-f002:**
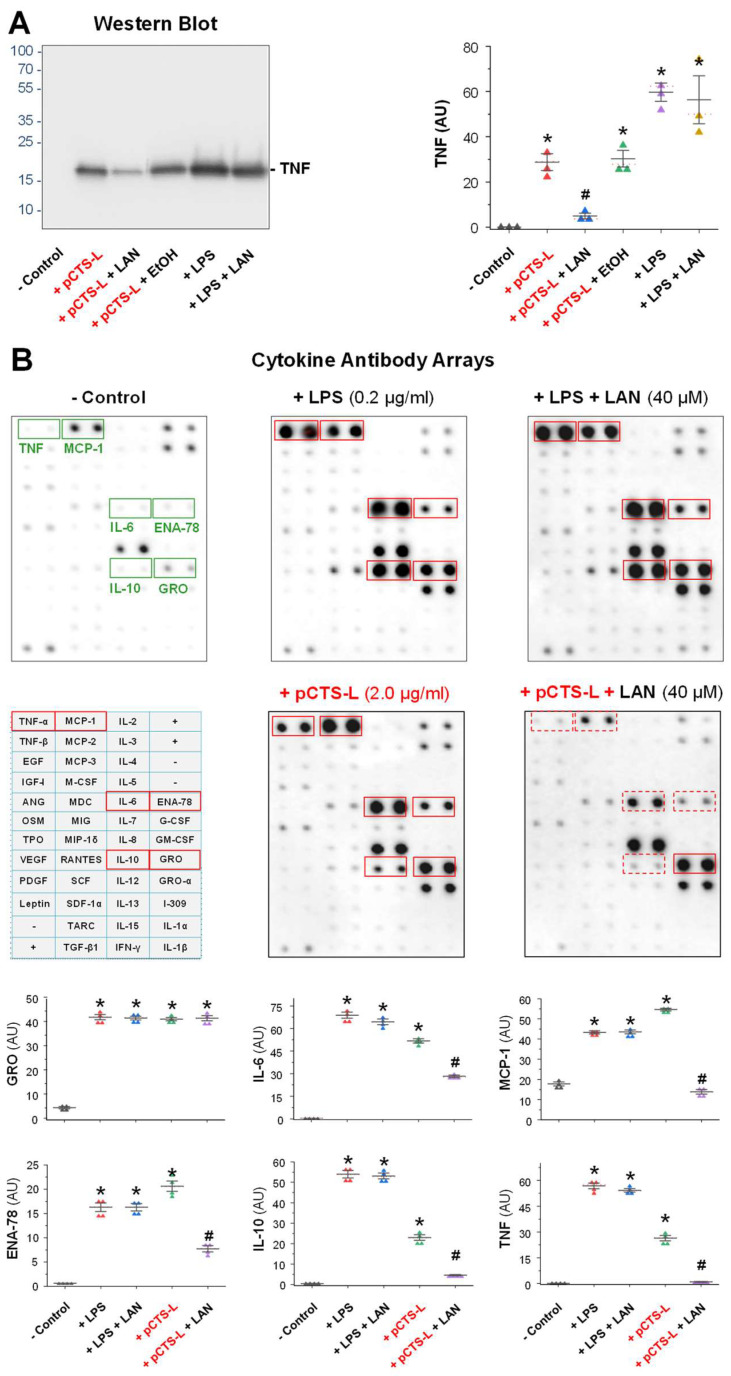
Lanosterol selectively attenuated pCTS-L-induced productions of cytokines/chemokines in human peripheral blood mononuclear cells (PBMCs). Human PBMCs were stimulated with LPS (0.2 μg/mL) or recombinant human pCTS-L (2.0 μg/mL) in the absence or presence of LAN (40 μM in Ethanol, EtOH) for 16 h. The levels of TNF and other cytokines and chemokines in cell-conditioned medium were determined using Western blotting (Panel (**A**)) or Cytokine Antibody Arrays (Panel (**B**)) and expressed as arbitrary units (AU). Note that the release of TNF and several other cytokines/chemokines induced by pCTS-L, but not LPS, was significantly inhibited by LAN. *, *p* < 0.05 versus “−pCTS-L”; #, *p* < 0.05 versus “+pCTS-L”, one-way ANOVA.

**Figure 3 ijms-24-08649-f003:**
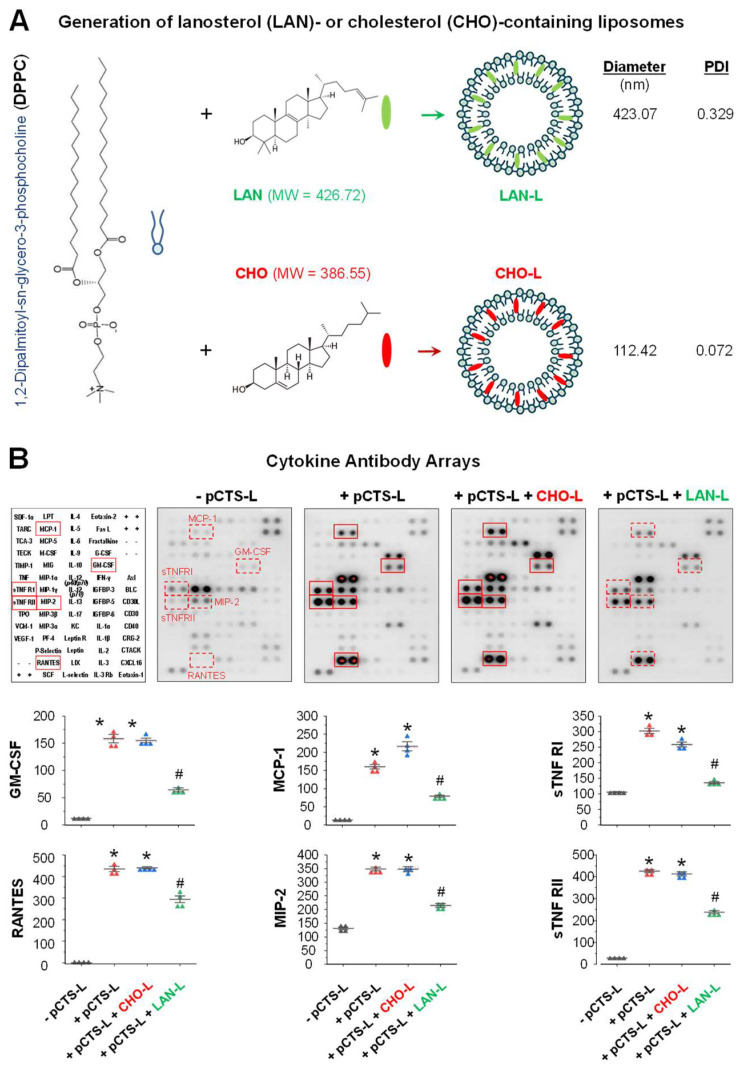
DPPC-based liposomes carrying lanosterol (LAN-L), but not cholesterol (CHO-L), significantly attenuated pCTS-L-induced production of cytokines and chemokines in murine macrophages. (**A**) Scheme for the generation of DPPC-based liposomes containing either lanosterol (LAN-L) or cholesterol (CHO-L). Given their amphiphilic nature, both LAN and CHO can easily be intercalated into DPPC-based lipid-bilayer with the hydrophobic steroid skeleton embedded in the bilayer core while the hydroxyl group is positioned at the hydrophobic-hydrophilic interface. (**B**) LAN-L effectively inhibited pCTS-L-induced cytokines and chemokines in murine macrophages. Murine macrophage-like RAW 264.7 cells were stimulated with recombinant murine pCTS-L in the absence or presence of LAN-L (40 μM) or CHO-L (40 μM) for 16 h. The extracellular concentrations of cytokines and chemokines were determined using Cytokine Antibody Arrays and expressed as arbitrary units (AU). Note that pCTS-L induced the release of several cytokines and chemokines, which were significantly inhibited by DPPC-based liposomes containing LAN (LAN-L), but not CHO (CHO-L). *, *p* < 0.05 versus “−pCTS-L”; #, *p* < 0.05 versus “+pCTS-L”, one-way ANOVA.

**Figure 4 ijms-24-08649-f004:**
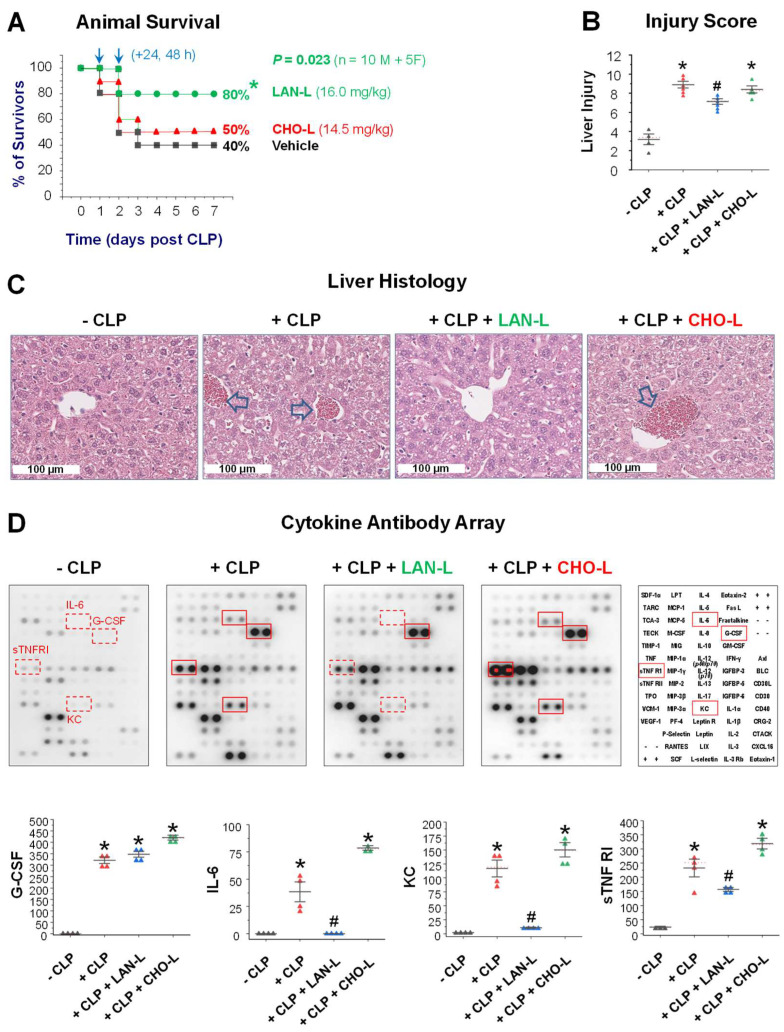
LAN-containing liposomes rescued mice from lethal sepsis partly by attenuating sepsis-induced injury and systemic inflammation. (**A**) LAN-containing liposomes (LAN-L) rescued mice from lethal sepsis. Male (M) and female (F) Balb/C mice were subjected to CLP, and LAN-L or CHO-L was given intraperitoneally at indicated doses and time points. Animal survival rates were monitored for two weeks to ensure no later death. *, *p* < 0.05 versus saline control group. (**B**,**C**) LAN-containing liposomes (LAN-L) attenuated sepsis-induced liver injury. Balb/C mice were subjected to CLP, and LAN-L or CHO-L was given intraperitoneally at 2 h and 20 h post CLP, and liver was harvested at 24 h post CLP for H&E staining and histological analysis. Liver injury scores were expressed as means ± SEM of 4–6 animals per group. *, *p* < 0.05 versus “−CLP”; #, *p* < 0.05 versus “+CLP”. (**D**) LAN-L significantly attenuated sepsis-induced systemic inflammation. Balb/C mice were subjected to CLP, and CHO-L (14.5 mg/kg) or LAN-L (16.0 mg/kg) was given twice at 2 h and 20 h post CLP. At 24 h post CLP, animals were sacrificed to harvest blood and measure various cytokines and chemokines and expressed as arbitrary units (AU). *, *p* < 0.05 versus CLP”; #, *p* < 0.05 versus “+CLP”.

**Figure 5 ijms-24-08649-f005:**
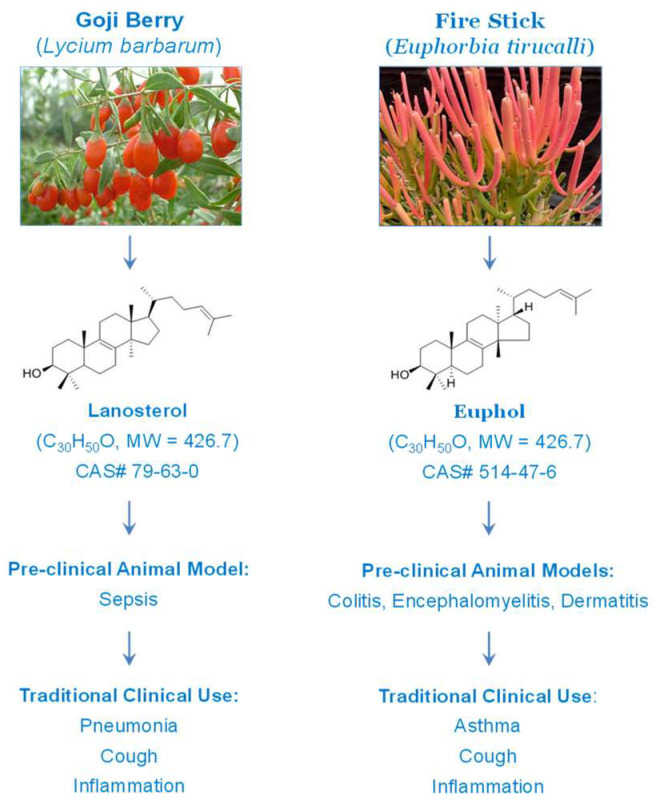
Protective effects of Goji Berry-derived lanosterol and Fire Stick-derived euphol in animal models of inflammatory diseases. Although lanosterol and euphol contain an identical steroid skeleton structure, they carry methyl groups with different orientations in three-dimensional space and are thus termed as “cis-trans isomers”. The therapeutic efficacy of these sterols supports the traditional use of these plants in the clinical management of various inflammatory ailments.

## Data Availability

All research data are presented in the figure section of this research article.

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
