# Peer review of "Development of Procathepsin L (pCTS-L)-Inhibiting Lanosterol-Carrying Liposome Nanoparticles to Treat Lethal Sepsis"

_ijms, 2023, doi:10.3390/ijms24108649_

Round 1

Reviewer 1 Report

In the article 'Development of procathepsin L (pCTS-L)-inhibiting lanosterol-2 carrying liposome nanoparticles to treat lethal sepsis', submitted for review, the authors conducted research towards developing a high-throughput screening assay to search for small molecule inhibitors of pCTS-L-mediated inflammation in order to develop novel therapies for lethal sepsis. They demonstrated that the natural lipophilic sterol, lanosterol (LAN) can be a selective inhibitor of pCTS-L-induced inflammation. They developed LAN-carrying liposome nanoparticles as a perspective therapy for lethal sepsis. To my knowledge, the authors have carried out innovative research in the development of new therapies for sepsis.

The manuscript is clearly written and presented in a well-structured manner. The cited references are mostly recent publications and relevant. Figures are clear, developed with great care. The experimental design is appropriate to test the hypothesis. The data are interpreted appropriately and consistently throughout the manuscript. The conclusions are consistent with the evidence and arguments presented.

I only suggest conducting stability studies for the prepared liposomes, in accordance with the ICH guidelines (Harmonised Tripartite Guidelines).

Author Response

We thank the reviewer for the thorough review and these positive comments.  Although we purposely used phospholipids with long and saturated acyl chains to produce liposomes with relatively higher stability, liposomes commonly suffer from storage instability caused by particle fusion and aggregation, as well as phospholipid hydrolysis or oxidation.  That was why as soon as the LAN- or CHO-carrying liposomes were generated, we immediately conducted all cellular and animal experiments within 1-6 weeks, long before subsequent appearance of precipitates (possibly due to liposome self-aggregation, coalescence and flocculation) after 6 – 9 months of prolonged storage (4°C).  Therefore, stability studies for the lanosterol-carrying liposomes are critically needed before conducting future clinical trials in accordance with quality guidelines of the International Council for Harmonisation of Technical Requirements for Pharmaceuticals for Human Use (ICH)  (Lines 276-286).   

Reviewer 2 Report

Dear Authors

Greetings

Finally, a few minor comments are provided. Overall, this manuscript is written exceptionally well. The use of English language is very good, and the sections follow clearly. In general, the text could be shortened. This work demonstrates the big potential, which is of great value for numerous applications.

Kind regards

Dear Authors

Greetings

Finally, a few minor comments are provided. Overall, this manuscript is written exceptionally well. The use of English language is very good, and the sections follow clearly. In general, the text could be shortened. This work demonstrates the big potential, which is of great value for numerous applications.

Kind regards

Author Response

We thank the reviewer for the positive comments, but respectfully request the reviewer’s understanding that it is rather difficult to further shorten this relatively concise and information-dense manuscript. 

Reviewer 3 Report

Thanks for this study. Potential new therapies for sepsis are urgently needed in light of today's rise in antimicrobial resistance, and limited armamentarium of immunosuppressive therapies. Therefore, I would like to thank you for doing this study. However, as with all preclinical and animal studies, there is a long road to go before patients will potentially benefit from this. The limitations of your findings are well outlined in the discussion section. Yet, I think it would be good to add a (short) section about next steps in clinical development, for the interested clinician who is likely less familiar with this type of fundamental research.

No concerns.

Author Response

We appreciate the reviewer’s comments, and have added a few sentences to describe next steps to improve the stability of liposomes before clinical testing of the lanosterol-carrying liposome technology (please see Lines 276-303 of the revised manuscript).